# Exploring support for medicalized female genital mutilation/cutting: A study on migrant women living in Italy

**Livia Elisa Ortensi**[ID][1]*, **Patrizia Farina**[2], **Daniela Carrillo**[ID][2], **Enrico Ripamonti**[3]

1 Department of Statistical Sciences "Paolo Fortunati", Alma Mater Studiorum University of Bologna, Bologna, Italy, 2 Department of Sociology and Social Research, University of Milan-Bicocca, Milan, Italy, 3 Department of Economics and Management, University of Brescia, Brescia, Italy

* livia.ortensi@unibo.it

## Abstract

The medicalization of female genital mutilation/cutting (FGM/C) as a harm reduction strategy is a highly debated issue, although largely unexplored among migrants living outside practising countries. This study investigates the extent of the support for FGM/C conditioned on its medicalization among migrant women from FGM/C-practising countries residing in Italy, and the characteristics of women supporting the practice. Data are from a national survey on FGM/C conducted in Italy in 2016, covering a representative sample of 1,378 women aged 18+ who were born in Nigeria, Egypt, Eritrea, Senegal, Burkina Faso, Somalia, and Ivory Coast. A discrete choice framework and a multinomial probit choice model are adopted to analyze women's preferences about FGM/C continuation and medicalization. Findings indicate that, compared with women who support the practice unconditionally, the requirement of medicalization correlates with higher educational level, age, being in a couple, and being from a country where FGM/C is more commonly medicalized. Perceived benefits linked to increased support for FGM/C medicalization include religious approval, better marriage prospects, cleanliness, and conformity to traditional cultural values. Our data show that higher education is a critical, but not unique, factor in understanding the support for FGM/C in its medicalized form.

## Introduction

The World Health Organization categorizes all procedures involving the partial or total removal of external female genitalia or any injury to the female genital organs for non-medical reasons under the umbrella term Female Genital Mutilation/Cutting (FGM/C) [1]. Although the exact number of girls and women worldwide who have undergone FGM/C remains unknown, in the 31 countries with representative data on prevalence, at least 230 million girls and women have been cut, and 3 million girls are still at risk of being subjected to this practice in all its forms [2]. FGM/C

**Data availability statement:** Data are available from the University of Milan Bicocca for researchers who meet the criteria for access to confidential data. Contact details: Department of Sociology and Social Research, University of Bologna Management secretariat segreteriadmin.sociologia@unimib.it

**Funding:** This study received funding from the European Commission under the European Union's CERV programme CERV-2022-DAPHNE (DORA - Data integration for acknowledging risks and protecting children from violence PI Livia Elisa Ortensi) Grant Agreement N. 101095969. https://ec.europa.eu/info/funding-tenders/opportunities/portal/screen/home https://ares20.it/dora/ LO is the PI of the project PF is the local PI for the University of Milan Bicocca ER is the local PI for the University of Brescia The funder did not play any role in the study design, data collection and analysis, decision to publish, or preparation of the manuscript.

**Competing interests:** The authors have declared that no competing interests exist.

prevalence is documented among women in African countries, with rates ranging from 1% in Cameroon and 4% in Ghana and Togo to 91% in Egypt and 98% in Somalia. Additionally, due to the migration of an increasing number of women from regions with high FGM/C prevalence, it has been estimated that at least half a million migrant women who have undergone FGM/C lived in Europe as of 2011 [3].

Although FGM/C is internationally recognized as a harmful practice and a human rights violation, it is increasingly being medicalized as a harm reduction strategy to minimize its adverse effects on health. In many countries where FGM/C is traditionally practiced, the prevalence rates of medicalization are rising [4].

Initially, campaigns against female genital mutilation/cutting mainly focused on highlighting the negative health consequences of the practice to raise awareness and encourage people to abandon it. This approach was valid in terms of increasing awareness of the health risks associated with the practice [5]. However, many experts have recognized that it also has unintentionally led to the medicalization of FGM/C rather than to its eradication [1]. A recent overview based on data from 2006 to 2017 for 24 countries found that 18% of cut girls under age 15 had the practice performed by a healthcare provider, as reported by their mothers. The proportion is around 70–80% in Egypt and Sudan, between 20 and 30% in Kenya, Djibouti and Guinea, and between 10 and 30% in Nigeria, Iraq, and Yemen. 93% of women who underwent medicalized FGM/C live in Egypt, Nigeria, and Sudan, with over half residing in Egypt [6]. Elsewhere, medicalized cutting is rare and restricted to geographically defined areas [7].

FGM/C can never be "safe," and there is no medical justification for the practice. FGM/C has irreparable physical and psychological consequences and violates the right to health, freedom from violence, bodily integrity, non-discrimination, and freedom from cruel, inhuman, or degrading treatment [6]. Even when performed by a healthcare provider in a sterile environment, there are immediate and long-term health risks [6]. Reports indicate that due to sedation, the amount of flesh cut can be more than what is typically removed during traditional practices, worsening the effects [8]. Notably, the *savoir-faire* of the practice has rarely been institutionalized, with few exceptions, such as in Egypt for a short period. The technique is often taught by older colleagues who learned it as it is traditionally transmitted through healers [9]. In general, FGM/C is more likely to be practiced by doctors who are less informed about health *sequelae* [10].

The medicalization of FGM/C among migrants living in European countries and the USA has been documented [4]. However, to date, no studies have analyzed these issues using quantitative methodologies. To address this gap, the present investigation aims to understand the reasons for supporting the practice and the background characteristics of immigrant women (from countries where FGM/C is practiced) living in Italy who support the continuation of FGM/C under the condition of its medicalization, compared to women who unconditionally support the practice and those who oppose its continuation. Our study is based on a survey carried out in Italy in 2016 within the framework of the DAPHNE project FGM-Prev, funded by the European Commission. The survey involved 1,378 women aged 18+born in Nigeria,

Egypt, Eritrea, Senegal, Burkina Faso, Somalia, and Ivory Coast. A discrete choice framework and a multinomial probit choice model are adopted to analyze women's preferences regarding FGM/C continuation and medicalization.

The study put forward three main research questions (**RQs**). First, it aims to understand the extent of support and, consequently, the potential demand for medicalization among migrant women coming from the most relevant FGM/C-practising countries in Italy, despite the practice being illegal (**RQ1**). There is currently no data from any European country on migrant women's support for FGM/C medicalization. Second, the study aims to assess whether women who support FGM/C only under the condition of medicalization have different background characteristics compared to (i) women who unconditionally support the practice and (ii) those who do not support it (**RQ2**). Third, the study aims to assess the relevance of the perceived benefits of FGM/C in supporting its medicalization (**RQ3**).

The findings of our study are invaluable for informing policymakers and designing campaigns to raise awareness about the continued practice of FGM/C, even in its medicalized form.

## Theoretical background

### Defining FGM/C and its medicalization

According to the WHO classification, four types of FGM/C exist, each with subtypes [11]. In general terms, types I and II are less invasive than type III and include clitoridectomy (type I) and excision of the labia (type II). The absence of a specific surgical procedure implies that different levels of harm can occur. Type III (infibulation) is the most severe form of the practice; the operation leaves a tiny orifice for menstrual blood and urine, and the suture must be reopened at the beginning of sexual life or during childbirth, causing further complications. The prevalence of this form of FGM/C has been decreasing over time [12]. Finally, the definition of FGM/C type IV is blurred, comprising pricking, scraping, nicking, and piercing.

Amid the many elements that have changed over the past decades, there has been a shift towards less invasive forms of the practice [13], along with an increased demand for its medicalization [14]. According to the WHO [1], *medicalization* refers to a "situation in which FGM/C is practiced by any category of healthcare provider, whether in a public or private clinic, at home, or elsewhere" (pag.2). However, the term has been used more broadly in the literature to include the provision of medical supplies for surgical procedures and the performance of operations in clinics or hospitals by trained healthcare providers [14]. The medical procedure also involves de-infibulation and re-infibulation, the latter being debated in Europe as infibulated women may request to return to their previous state after being unsewn due to childbirth or other medical procedures [15–16].

### Medicalization as a step towards FGM/C eradication?

Campaigns against FGM/C initially focused on the practice's health consequences to raise awareness and motivate its abandonment [4]. However, the current picture of FGM/C medicalization seems to represent a development of the phenomenon distinct from an intermediate phase towards FGM/C eradication [8]. Despite being a form of harm reduction, scholars' opinions about the medicalization of FGM/C are mixed. In a recent review of the literature, Shell-Duncan et al. [14] argue that there is no evidence to support either the idea that medicalization will lead to the suppression of the practice or that it will support its perpetuation. Consequently, Shell-Duncan has revised her previous stance in favor of medicalization as a form of harm reduction [17]. Anti-medicalization advocates claim that medicalization institutionalizes the practice and its performance by respected members of society, such as healthcare providers, implying tacit approval that makes it more arduous to contrast [18–19]. Moreover, the diminished risk of short-term consequences, considering there are no assurances on the long-lasting effects, can lead to a reduced awareness of the overall impact of the practice on health [20–22].

A concomitant interpretation highlights the valuable opportunities that medicalization provides for discussing the practice with health professionals. Egyptian women, particularly when doubtful, are reported to ask doctors for advice on the need to cut their daughters and pay great tribute to the fact that the advice comes after the physical exam [22].

Some scholars, criticizing Western paternalistic eradication campaigns, advocate for medicalization as a form of self-determination for adult women who choose to undergo the practice [23–25]. Onsongo, for instance, stresses the importance of allowing consenting women of age to decide on their own bodies, thus permitting medical-assisted procedures [26]. It should be clear that these critical positions do not in any way consider the cutting of underage girls and aim instead to create a parallel with Western cosmetic surgery. Shweder goes even further, affirming the importance that medical practice can have in maintaining a procedure that aligns with the social construction of identities and belonging [24]. However, FGM/C is typically performed on girls under the age of 14 who are minors, with no faculties to consent and are unlikely to have the capacity to resist or object to the practice or even fully understand its implications [27]. In 2008, the WHO, released a statement opposing the medicalization of FGM/C, stating that it not only condones and perpetuates a harmful practice but also constitutes a dangerous and criminal act [28]. In 2010, several UN agencies and international medical bodies developed a joint "Global strategy to stop healthcare providers from performing female genital mutilation", unequivocally advising that FGM/C in any form should not be performed by health professionals in any setting [1]. In the following years, this clear stance was supported by others, notably the Society of Paediatrics and of Gynaecologists and the World Medical Association [29–30]. The 2020 UN Human Rights Council resolution on the eradication of FGM/C recognizes medicalization as a significant challenge and warns against the practice.

### Parents' reasons for seeking FGM/C medicalization: Evidence from practising countries

Two main reasons for adopting medicalization can be recognized, connected to the ideas of safety and modernity. Parents seek medical circumcision because it is perceived as safer, particularly for short-term effects such as haemorrhage, infections, or shock [22]. Although there are no protocols on how to perform the surgery operation due to the universal stance against it [14], it is generally agreed that medical execution is safer [8]. Improved health services may *per se* also sustain the demand for medicalization, as Obianwu and colleagues found in Nigeria [9]. Concurrently, considering that the practice has been outlawed almost everywhere, limiting its short-term adverse consequences reduces the likelihood of being caught by the authorities [8].

The notion of modernity enhances the perception of what is considered safe and, consequently, acceptable. The recognized authority of the medical community, combined with the use of "modern" instruments and medications from Western medicine, helps to frame the practice as both novel and acceptable [8,20].

Who are the women who seek medicalized FGM/C for their children, and how do they differ from those women who do not question the practice? Evidence from practising countries suggests these women may have different socio-economic positions [31]. Medicalization requires financial resources, awareness of options, faith and confidence in Western medicine, and the freedom to travel, in some cases to private clinics that are distant or abroad - elements typically associated with upper-class women. This has been clearly described as happening in Egypt, where doctors mainly perform the practice [32]. In contrast, a different picture emerges from Nigerian areas, where FGM/C is still largely carried out and has been extensively researched [9,33]. Midwives or nurses execute medicalization: considering that FGM/C are frequently performed on girl babies in their first weeks, those who attend childbirth are usually called for the practice [20]. While the condition of modernity is maintained, medicalizing the practice in such a context does not require high economic costs or high awareness of its consequences. The medicalized procedure is more easily carried out in urban areas [5], where women are generally more exposed to different inputs regarding what and how to perform it. Similarly, in Indonesia, medicalized FGM/C is performed as part of the package of services for newborns in health facilities [6].

In summary, the contemporary scenario is highly varied but in general, higher levels of education and wealth are associated with increased requests for medicalization [22,31,34]. This contradicts the common belief that education will counteract the practice; instead, it seems to transform it. As Morhason-Bello et al. have stated [34], despite the general belief that women are susceptible to FGM/C due to poor education and poverty, our study found that educated women can also be affected by seeking the practice in its less harmful form.

## Support for FGM/C and its medicalization among migrant women: An understudied issue

Understanding why many women continue to support and value FGM/C despite years of campaigns and funding aimed at eliminating it is challenging. Various spheres, such as religion, cultural and social identity, womanhood, and patriarchy, are alternatively or simultaneously invoked [11]. FGM/C practice is framed as a social norm or convention [35–37]. Individuals conform to a particular rule of behavior because they expect that a significant portion of their social or reference group (such as ethnic or religious) will also conform. Compliance with the rule is considered to be in an individual's best interest in the case of a social convention, whereas, in the case of a social norm, adherence to a given practice is motivated by both social rewards and fear of social penalties for non-compliance. However, further analysis has shown that a more revealing picture emerges when structural factors are also taken into account. [38].

Mass migration has further complicated this picture. To date, FGM/C is present in many migration countries not customarily touched by it, spanning from Europe to Australia, North America, the Middle East, and North Africa [39], brought by immigrant communities. The migration process is a factor potentially contributing to the abandonment of the practice in immigrant communities or the transition to adopting different and less invasive forms [40–41]. Migrant women and men are pivotal in this process, and exposure to different narratives regarding gender relationships, religious dictates, and marital practices plays a fundamental role in its abandonment [42–43].

Conversely, the feeling of uprooting and the challenges connected to integration processes may reinforce the attachment to the practice, seen as valuable for maintaining traditions and raising girls in what is perceived as an unsafe context [13,44]. This phenomenon is increasingly observed in refugee camps, where, in addition to identity reasons, the role of protecting women from sexual intercourse gains even greater significance [13,40]. Moreover, many women who have undergone FGM/C and now live abroad may view cut genitalia as beautiful and clean, perceiving the practice as symbolically significant and a means to enhance their femininity [45].

However, in Western countries, the practice is highly condemned socially and legally. Health services, not entirely equipped for dealing with it, are primarily focused on prevention and/or reparation surgeries. Much research, mainly qualitative, has explored the attitudes towards continuing or abandoning the practice *tout-court* [46–47]. However, to the best of our knowledge, no studies have investigated support for medicalization. As mentioned earlier, literature covering migrants' countries of origin has largely depicted an increasing request for such procedures. Given the intertwined material and symbolic relationship and dependency between migrants and their places of origin, medicalization in Europe cannot be ignored. Ali et al. shed light on the subject of FGM/C and its medicalization among migrant adolescents residing in the UK [48]. While the article highlights the existence of positive attitudes towards the practice, it is essential to note that the findings are not necessarily representative of the entire population. For some informants, performing the procedure in a sanitary environment provides it with the appeal of safety and modernity [48]. Without the traits of archaicity and riskiness, FGM/C would then maintain its value in preserving the sense of identity, which is highly advocated in receiving contexts often perceived as hostile.

## Female genital mutilation/cutting among migrant women and children in Italy

Italy has become a significant immigration country since the 1990s. The gradual feminization of the migrant population, driven by family reunification and independent female migration (notably from Nigeria), has led to a substantial number of women from FGM/C-practising countries. As of January 1st, 2022, approximately 194,000 women born in an FGM/C-practising country were living in Italy [ISTAT, 2023]. This number has increased by 25.3% over the past decade. The most prominent communities are from Egypt (43,800), Nigeria (42,300), and Senegal (27,700). Previous research conducted in Italy on the intergenerational transmission of FGM/C to daughters shows that certain factors correlate with the risk of being cut for second-generation girls. These factors include having a mother who experienced FGM/C, being born before migration to Italy, and low maternal empowerment (measured in terms of education and attitudes toward intimate partner violence) [44]. According to a study partially based on the same data source of the present investigation, 60–80 thousand

foreign-born women aged 15 and over with FGM/C were living in Italy in 2016 [49]. A study conducted in Italy by EIGE estimated that, in 2016, 15% to 24% of girls were at risk of FGM/C in Italy out of a total population of 76,040 girls aged 0–18 originating from countries where female genital mutilation/cutting is practiced [50]. Given the high and rising number of girls potentially at risk of FGM/C, it is vital to understand the characteristics of mothers who may seek medicalization of FGM/C for their daughters.

## Methods and Materials

### Data

Data comes from the 2016 national survey on Female Genital Mutilation conducted in Italy as a part of the Daphne project FGM-Prev [51]. The study received ethical clearance from the ethical committee of the University of Milan-Bicocca (Ethical committee of the University of Milan – Bicocca; prot. 209 FARINA). Conducted from June to December 2016, the survey covered a representative sample of 1,378 women aged 18 and over, living in Italy, and born in Nigeria, Egypt, Eritrea, Senegal, Burkina Faso, Somalia, and the Ivory Coast. The survey methodology combined facility-based and respondent-driven sampling [51]. Women were interviewed in many Italian cities and towns, including suburban and mountain areas. The survey aimed to study FGM/C within the most relevant communities to assess the prevalence and risk of undergoing the practice among second-generation girls. The FGM/C status was self-reported by the women interviewed, with no physical examinations performed. The interviews were conducted by a team of female foreign interviewers from the targeted communities, who were well acquainted with the issue. Their ability to translate and formulate questions appropriately was crucial in facilitating intimate conversations and reducing voluntary underreporting [51]. Consent to participate in the study was obtained orally, and the questionnaire was fully anonymous.

Among other information collected, the survey asked respondents if they had ever heard of the practice and, if so, about their attitudes toward its continuation. Most women interviewed were aware of FGM/C. To ensure that the interviewee was genuinely not familiar with the practice, the question was asked twice with different wording. Only 57 women (4.1% of the unweighted sample) were unaware of FGM/C. Consequently, these women were excluded from the analysis.

### Methods

A discrete choice framework is adopted, specifically utilizing a multinomial probit choice model to analyse women's support for the continuation of FGM/C. This model extends the binary probit choice model to scenarios in which individuals choose among three or more alternatives—in our case, the three options regarding FGM/C continuation: stop, support, and support conditioned on medicalization. These outcomes are modeled by regression on alternative-specific (related to the practice of FGM/C) and case-specific (pertaining to women) covariates. Underlying the model are $j$ utilities

$$\eta_{ij} = x_{ij}\beta + z_i\alpha_j + \xi_{ij}$$

where $i$ denotes women and $j$ denotes the alternatives, with $i = 1, \ldots, N$ and $j = 1, 2, 3$. $x_{ij}$ is a 1×8 vector of alternative-specific variables, $\beta$ is a 8×1 vector of parameters, $z_i$ is a 1×10 vector of case-specific variables, $\alpha_j$ is a 10×1 vector of parameters for the $j$th alternative, and $\xi_{ij} = (\xi_{i1},..., \xi_{ij})$ is distributed as multivariate normal with mean zero and covariance matrix $\Omega$. The $i$-th woman selects the alternative whose utility $\eta_{ij}$ is highest.

The chosen model required us to reshape the original data in the long form.

The multinomial probit choice model is similar to McFadden's choice model (multinomial logit model) but differs in its assumption regarding the independence of irrelevant alternatives (IIA). Unlike McFadden's model, the multinomial probit model allows the covariance of $\xi_i$ to have a general structure, relaxing the IIA assumption. The IIA assumption is particularly strong, as it asserts that the relative probability of selecting one alternative should not change if another alternative is introduced or eliminated. In our case, this assumption is challenging to meet because our informants state that they support FGM/C only if medicalization is available. We, therefore, assume that if one alternative (particularly medicalization)

is not available, women would likely change their choice, with the direction is relatively unpredictable. The multinomial probit choice model addresses this by directly modelling the correlation between the error terms for different alternatives. Nevertheless, as a consistency check, we fitted a McFadden's choice model, and the results were fully consistent with the multinomial probit model presented here.

Women are naturally grouped into ten categories based on their citizenship at birth. To account for clustering within these groups, the standard errors of the estimated coefficients are adjusted using a sandwich estimator. This robust estimator calculates the coefficients' covariance matrix by taking into account the within-cluster correlation of the errors.

## Measures

The dependent variable is based on the question: "*Do you think that this practice [FGM/C] should continue*?". Possible answers were "*No*", "*Yes*", "*Yes but only if carried out in a hospital*", "*Unsure*", and "*Refuses to answer*". Women who agreed with the third option were therefore considered favorable to FGM/C conditionally on medicalization. For our analysis, women who declared unsure (57) or refused to answer (16) were excluded from the study as we cannot attribute any characteristic in terms of perceived FGM/C desirability to their choice. We exclude, for this reason, 6.9% of women who have ever heard of FGM/C. Thus, the final sample used for the model is composed of 1,248 women.

The study adopts a social norms and social conventions framework to evaluate utility [38,52]. In the multinomial probit choice model framework, perceived benefits (utility) of FGM/C are considered as alternative-specific characteristics that vary across both cases and alternatives and are common to supporting FGM/C either entirely or under the condition of medicalization.

Explanatory variables are reported in Table 1. The first set of variables refers to alternative-specific features describing the women's beliefs and adherence to social norms regarding FGM/C, whether in its medicalized form or not. The second set refers to women's socioeconomic and family characteristics that are case-specific variables, varying only across cases (i.e., women). Finally, the models also account for the diffusion of medicalization in the country of origin by controlling for the prevalence of FGM/C medicalization, using data retrieved from Shell-Duncan and colleagues [14].

## Modeling strategy

Table 2 shows our modeling strategy. Initially, a model was fitted on the full sample (Model 1a), followed by a second model fitted solely on cut women, taking women who support the practice unconditionally as a reference group (Model 2a). The decision to re-fit the model on a subsample of cut women was based on literature findings from both emigration and practicing countries [2,44], which indicate that uncut women rarely support the practice, whereas support is usually higher among cut women. Both models 1a and 2a designate women who support the practice unconditionally as a reference group. In the second step, both models were re-fitted considering women who do not support the practice as the reference group. Full results from these models are available in the Supporting Information files S1 and S2 Appendix.

## Results

### Descriptive analysis

The women's mean age is 34.6 years, and the mean age at migration is 23.9 years. 29% attained the primary educational level at most, 56.4% attained the secondary level, and 14.7% attained the tertiary level. 57.1% of the women are active in the labor market, 59.0% are in a relationship (living together or apart) at the time of the interview, and 50.8% are cut (Table 3).

Table 4 shows the sample composition by citizenship at birth and the support for medicalization within each group (**RQ1**). Although the small sample size at the citizenship of origin level cannot support robust conclusions, it is evident that support for medicalization is higher among women originating from countries where it is more prevalent (Nigeria

**Table 1. Variables included in the model.**

| Type of explanatory variable | Variables | Categories |
|---|---|---|
| Alternative specific | Perceived benefits of FGM/C: Cleanliness/hygiene | Yes; No, reference category |
| | Perceived benefits of FGM/C: Social acceptance | Yes; No, reference category |
| | Perceived benefits of FGM/C: Better marriage prospects | Yes; No, reference category |
| | Perceived benefits of FGM/C: Preserve virginity/Prevent premarital sex | Yes; No, reference category |
| | Perceived benefits of FGM/C: Preserve cultural traditions of parents/ancestors | Yes; No, reference category |
| | Perceived benefits of FGM/C: To instill discipline and traditional cultural values | Yes; No, reference category |
| | Perceived benefits of FGM/C: Religious approval | Yes; No, reference category |
| | Perceived benefits of FGM/C: More sexual pleasure for men | Yes; No, reference category |
| Case-specific (i.e., referring to women) | Age at migration | In years |
| | Age | In years |
| | Highest level of achieved formal education | None or Primary, reference; secondary; tertiary |
| | The woman is active in the labour market | Yes, i.e., employed/unemployed; No, reference |
| | Family status | Single/separated/widowed, reference; In a relationship living apart; In a relationship living together; Refuses to answer |
| | The woman is married to an Italian native | Yes; No, reference category |
| | The woman's mother is cut and lives in Italy | Yes; No, reference category |
| | The woman is cut | Yes; No, reference category |
| | The woman regularly returns to the country of origin (i.e., at least once in the last 3 years) | Yes; No, reference category |
| | Number of female daughters | in digits |
| Country of origin related | Prevalence of FGM/C medicalization in the country of origin | % |

**Table 2. Modelling strategy.**

| | Sample | Reference |
|---|---|---|
| Model1a | Full sample: all women except those unsure about FGM/C continuation | Women unconditionally supporting FGM/C |
| Model 2a | Only cut women: all cut women except those unsure about FGM/C continuation | Women unconditionally supporting FGM/C. |
| Model1b | Full sample: all women except those unsure about FGM/C continuation | Women not supporting FGM/C |
| Model 2b | Only cut women: all cut women except those unsure about FGM/C continuation | Women not supporting FGM/C |

and Egypt), as well as among women from Burkina Faso. Overall, the support for FGM/C continuation is not widespread (23%), with 38% of women supporting FGM/C only if the practice is medicalized.

When considering only women who have experienced FGM/C—and are therefore most likely to support the practice [43]—the support for FGM/C rises to 38.7%. Among those favorable to the practice, 38.9% condition its continuation on medicalization. According to our results, medicalization is not the predominant option among women supporting FGM/C, except for those from Egypt.

Table 5 reports the main findings obtained in the multinomial probit choice model. First, the study aims to understand the characteristics of women whose support for FGM/C is conditioned on medicalization compared to those who support the practice unconditionally (**RQ2**). Our models show that compared to women who support the practice unconditionally, support of FGM/C conditional to medicalization correlates with age (higher age associated with conditional support),

**Table 3. Descriptive analysis of the sample.**

| | | | Value | Standard Deviation | N |
|---|---|---|---|---|---|
| Alternative specific characteristics | Perceived benefits for girls with FGM/C | Cleanliness/hygiene (%) | 6.8 | | 91 |
| | | Social acceptance (%) | 8.3 | | 110 |
| | | Better marriage prospects (%) | 12.1 | | 162 |
| | | Preserve virginity/Prevent premarital sex (%) | 17.6 | | 234 |
| | | Preserve cultural traditions of parents/ancestors (%) | 16.3 | | 215 |
| | | More sexual pleasure for the man (%) | 6.7 | | 86 |
| | | To instill discipline and traditional cultural values (%) | 9.0 | | 118 |
| | | Religious approval (%) | 5.1 | | 71 |
| Case (women's) specific characteristics | Age at migration (mean) | | 23.9 | 8.777 | |
| | Age (mean) | | 34.6 | 11.597 | |
| | % Highest education: primary or none | | 29.0 | | 375 |
| | % Highest education: secondary | | 56.4 | | 776 |
| | % Highest education: tertiary | | 14.7 | | 200 |
| | % Active | | 57.1 | | 787 |
| | % Single/separated/widowed | | 36.9 | | 504 |
| | % In a relationship living apart | | 10.5 | | 143 |
| | % In a relationship living together | | 48.5 | | 662 |
| | % with an Italian native husband | | 6.5 | | 51 |
| | % with a cut mother living in Italy | | 5.1 | | 70 |
| | % cut | | 50.8 | | 796 |
| | % regularly returns to her country of birth | | 46.8 | | |
| | Number of female daughters (mean) | | 0.7 | 0.891 | |

**Table 4. Sample composition and support for FGM/C according to citizenship at birth.**

| | Sample | | | Support for FGM/C continuation | | | |
|---|---|---|---|---|---|---|---|
| Country of origin | N | % by country on the total sample | % cut by country | No | Only if carried out in a hospital | Yes | Total |
| Cameroon | 64 | 4.7 | 16.9 | 89.1 | 3.1 | 7.8 | 100.0 |
| Ethiopia | 17 | 1.5 | 15.8 | 94.1 | 0.00 | 5.9 | 100.0 |
| Eritrea | 134 | 9.7 | 48.5 | 98.5 | 1.5 | 0.00 | 100.0 |
| Ghana | 50 | 3.9 | 7.4 | 96.0 | 4.0 | 0.0 | 100.0 |
| Cote d'Ivoire | 103 | 8.4 | 7.8 | 97.1 | 0.0 | 2.9 | 100.0 |
| Nigeria | 155 | 12.1 | 74.9 | 38.7 | 13.6 | 47.7 | 100.0 |
| Senegal | 133 | 10.3 | 29.8 | 88.7 | 9.0 | 2.3 | 100.0 |
| Somalia | 127 | 9.5 | 81.7 | 96.9 | 0.8 | 2.4 | 100.0 |
| Egypt | 241 | 22.2 | 52.9 | 78.0 | 17.8 | 4.2 | 100.0 |
| Burkina Faso | 224 | 17.7 | 70.1 | 53.1 | 11.6 | 35.3 | 100.0 |
| Total | 1248 | 100 | 43.6 | 77.0 | 8.7 | 14.3 | 100.0 |
| Total among women with FGM/C | – | – | – | 61.3 | 15.0 | 23.6 | 100.0 |

education (higher education associated with conditional support), family status, number of daughters (more daughters, less support), and prevalence in the country of origin (Table 5).

9. Second, when considering women who do not support the practice as a reference, variables correlated to differences in their support include age, age at migration, being cut, the number of female daughters, and the prevalence of

**Table 5. Multinomial probit choice models. Case-specific coefficients and robust standard errors from Model 1a and Model 2a for women who condition FGM/C continuation on medicalization vs women who support FGM/C unconditionally.**

| Women's specific variables - Choice: Support for FGM/C under the condition of medicalization vs unconditioned support | Model 1a Full Sample | | | Model 2a: Cut women only | | |
|---|---|---|---|---|---|---|
| | Coefficient | Robust standard error | P>\|z\| | Coefficient | Robust standard error | P>\|z\| |
| Age at migration | -0.037 | 0.027 | 0.177 | -0.031 | 0.027 | 0.242 |
| Age at the survey | 0.072 | 0.026 | 0.006 | 0.069 | 0.025 | 0.006 |
| Higher level of achieved formal education: secondary (ref. None or Primary) | 0.631 | 0.160 | <0.001 | 0.674 | 0.158 | <0.001 |
| Higher level of achieved formal education: tertiary (ref. None or Primary) | 0.788 | 0.261 | 0.002 | 0.705 | 0.262 | 0.007 |
| The woman is active in the labour market: Yes (ref. No, reference) | -0.129 | 0.208 | 0.534 | 0.005 | 0.213 | 0.982 |
| Family status: In a relationship living apart (ref. Single/separated/widowed) | 1.107 | 0.191 | <0.001 | 0.988 | 0.160 | <0.001 |
| Family status: In a relationship living together (ref. Single/separated/widowed) | 0.483 | 0.291 | 0.097 | 0.634 | 0.314 | 0.043 |
| The woman is married to an Italian native: Yes (ref. No, reference) | -1.094 | 0.803 | 0.173 | -1.381 | 0.886 | 0.119 |
| The woman is cut: Yes (ref. No, reference) | 0.737 | 0.504 | 0.143 | – | – | – |
| The woman regularly returns to the country of origin: Yes (ref. No, reference) | -0.029 | 0.239 | 0.902 | -0.089 | 0.208 | 0.669 |
| Number of female daughters | -0.301 | 0.104 | 0.004 | -0.291 | 0.102 | 0.004 |
| Prevalence of medicalization in the country of origin | 0.022 | 0.009 | 0.015 | 0.017 | 0.009 | 0.060 |
| Constant | -3.005 | 0.514 | <0.001 | -2.330 | 0.532 | <0.001 |

Note: Full models 1a and 2a are shown in in the Supporting Information file S1 Appendix. Akaike's information criterion for Model 1a: 776.398. Akaike's information criterion for Model 2a: 621.241.

medicalization in the country of origin (Table 6). Compared to women who do not support the practice, those who condition their support on medicalization are younger but have migrated at an older age.

2 These results can be read in terms of predicted probabilities according to various relevant variables. Concerning education, non-support of FGM/C tends to increase with higher education levels, whereas unconditional support declines. A different trend emerges with medicalization. Support for medicalization is more prevalent among women with secondary and tertiary education compared to those with only primary education. However, there is no significant difference in support levels between women with secondary and tertiary education (see Fig 1).

The support also tends to grow with age and to decrease among women with one or more daughters (Figs 2 and 3). Moreover, as expected, support for medicalized FGM/C is higher among women from countries where a larger proportion of women have undergone medicalized FGM/C.

As a further point of interest, perceived benefits statistically correlated with broader support for medicalization are investigated (**RQ3**). The model suggests that among the benefits, cleanliness/hygiene, better marriage prospects, preserving virginity or preventing premarital sex, preserving cultural traditions of parents/ancestors, instilling discipline and traditional cultural values and religious approval are related to increased support for medicalization. Table 7 shows changes in the predicted probabilities of supporting FGM/C either under the condition of medicalization or under no conditions, according to the women's perception of benefits for cut girls. The predicted probability of support for medicalization tends to grow more when women think FGM/C is approved by religion, if it entails better marriage prospects, and is perceived to improve cleanliness and instill discipline and traditional cultural values.

## Discussion

This study addresses the complex issue of supporting the practice of medicalized FGM/C among migrant women. To investigate the correlates of this phenomenon, the study relied on a sample of women residing in Italy, originating from Nigeria, Egypt, Eritrea, Senegal, Burkina Faso, Somalia, and Ivory Coast. While our findings are to be contextualized in the Italian scenario, it is worth highlighting that there are no similar studies from other European countries. Therefore, by

**Table 6. Multinomial probit choice models. Case-specific coefficients and robust standard errors from Model 1b and Model 2b for women who conditioning FGM/C continuation on medicalization vs women who support FGM/C unconditionally.**

| Women's specific variables - Choice: Support for FGM/C under the condition of medicalization vs no support | Model 1a Full Sample | | | Model 2a: Cut women only | | |
|---|---|---|---|---|---|---|
| | Coefficient | Robust standard error | P>\|z\| | Coefficient | Robust standard error | P>\|z\| |
| Age at migration | 0.061 | 0.013 | <0.001 | 0.031 | 0.013 | 0.015 |
| Age at the survey | -0.035 | 0.007 | <0.001 | -0.033 | 0.010 | 0.001 |
| Higher level of achieved formal education: secondary (ref. None or Primary) | 0.036 | 0.272 | 0.895 | -0.136 | 0.230 | 0.556 |
| Higher level of achieved formal education: tertiary (ref. None or Primary) | -0.403 | 0.282 | 0.153 | -0.590 | 0.296 | 0.047 |
| The woman is active in the labour market: Yes (ref. No, reference) | -0.169 | 0.253 | 0.505 | -0.648 | 0.297 | 0.029 |
| Family status: In a relationship living apart (ref. Single/separated/widowed) | 0.740 | 0.566 | 0.192 | 0.812 | 0.585 | 0.165 |
| Family status: In a relationship living together (ref. Single/separated/widowed) | 0.292 | 0.560 | 0.603 | -0.273 | 0.565 | 0.629 |
| The woman is married to an Italian native: Yes (ref. No, reference) | 0.671 | 0.607 | 0.269 | 1.222 | 0.648 | 0.059 |
| The woman is cut: Yes (ref. No, reference) | 1.482 | 0.468 | 0.002 | – | – | – |
| The woman regularly returns to the country of origin: Yes (ref. No, reference) | -0.299 | 0.228 | 0.191 | -0.329 | 0.235 | 0.160 |
| Number of female daughters | -0.347 | 0.134 | 0.009 | -0.339 | 0.132 | 0.010 |
| Prevalence of medicalization in the country of origin | 0.034 | 0.006 | <0.001 | 0.032 | 0.009 | <0.001 |
| Constant | -4.434 | 0.913 | <0.001 | -1.687 | 0.854 | 0.048 |

Note: Full models 1b and 2b are shown in in the Supporting Information file S2 Appendix Akaike's information criterion for Model 1b: 778.072 Akaike's information criterion for Model 2b: 623.140.

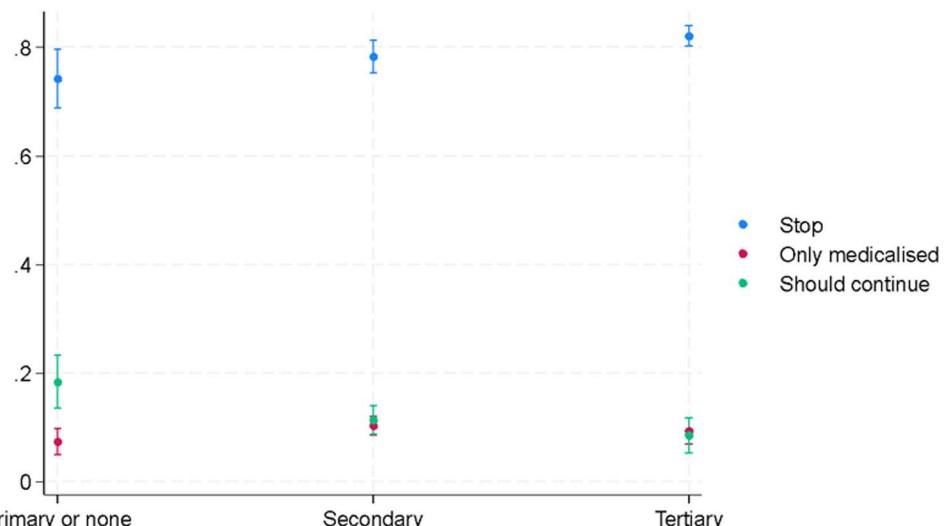

**Fig 1. Predicted probabilities of supporting the continuation of FGM/C by the highest level of education achieved (Model 1a).**

exploring the demand for medicalization among migrants settled in a Western country, the present results offer a unique insight into the broader discourse surrounding this practice.

Female genital mutilation/cutting has been the target of many enforcement initiatives that find legitimacy from the 2030 Agenda and the Istanbul Convention. Within this framework, medicalization is indeed a controversial issue and has been met with significant ethical and moral dilemmas [7,47]. The WHO and other prominent medical organizations oppose this practice due to its perpetuation of harm and potential legal ramifications [11]. Nevertheless, it

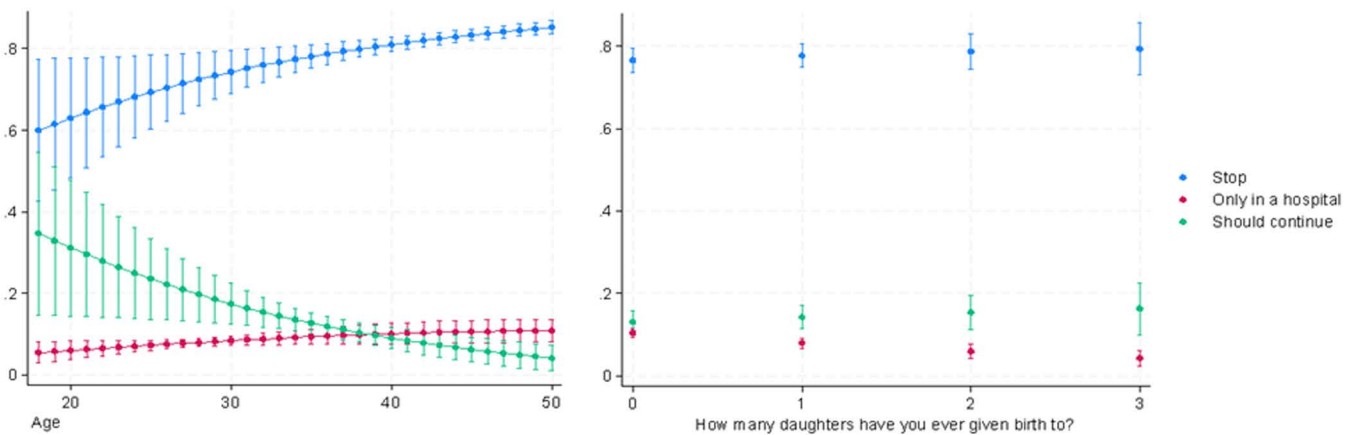

**Fig 2. Predicted probabilities of supporting the continuation of FGM/C by age at the time of the survey and the number of daughters ever born (Model 1a).**

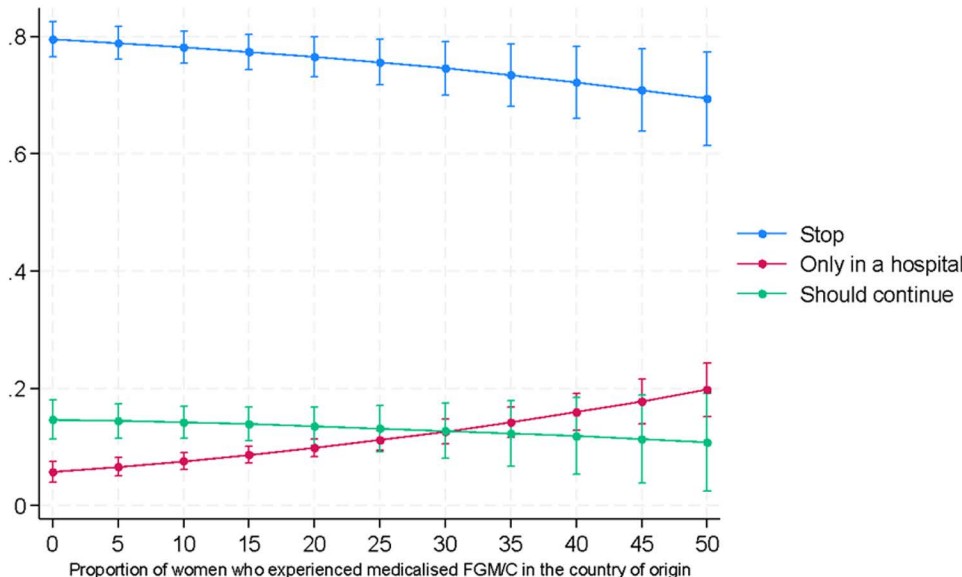

**Fig 3. Predicted probabilities of supporting the continuation of FGM/C by the level of medicalization in the country of origin (Model 1a).**

remains a subject of diverse opinions and attitudes within communities where FGM/C is prevalent, as described above [8,9,20,34]. Over time, the backdrop of FGM/C has undergone significant transformations. This includes a shift towards less invasive forms, often coupled with an increased inclination towards medicalization [6]. This evolution has contributed to the perpetuation of the practice while partially reducing harm in response to a range of perceived benefits, such as hygiene, social acceptance, better marriage prospects, and cultural preservation. While there are studies on the diffusion of medicalization in practicing countries, the present study calls attention to the lack of empirical evidence on the support of medicalized FGM/C among migrant women living in Western countries. Findings reveal a non-negligible percentage of women who support this form, particularly among women from Egypt, Nigeria, and Burkina Faso, for which the value is higher than 10% **(RQ1)**. While this endorsement might be expected among women originating from

**Table 7. Predicted probabilities of supporting FGM/C based on perceived benefits for cut girls (significant variables according to Model 1).**

| Perceived benefits of FGM/C: | Support conditioned on medicalization | | | | | Unconditioned support | | | |
|---|---|---|---|---|---|---|---|---|---|
| | No | overall predicted prob | Yes | Variation in predicted prob | P>\|z\| | No | overall predicted prob | Yes | P>\|z\| |
| Cleanliness/hygiene | 0.067 | 0.085 | 0.196 | 0.129 | <0.001 | 0.120 | 0.139 | 0.252 | <0.001 |
| Better marriage prospects | 0.056 | 0.085 | 0.219 | 0.163 | <0.001 | 0.090 | 0.139 | 0.283 | <0.001 |
| Preserve virginity/Prevent premarital sex | 0.051 | 0.085 | 0.171 | 0.12 | <0.001 | 0.084 | 0.139 | 0.140 | <0.001 |
| Preserve cultural traditions of parents/ancestors | 0.067 | 0.085 | 0.126 | 0.059 | <0.001 | 0.114 | 0.139 | 0.185 | <0.001 |
| To instill discipline and traditional cultural values | 0.063 | 0.085 | 0.191 | 0.128 | <0.001 | 0.110 | 0.139 | 0.249 | <0.001 |
| Religious approval | 0.064 | 0.085 | 0.326 | 0.262 | <0.001 | 0.121 | 0.139 | 0.373 | <0.001 |

countries where medicalization is more prevalent, our data suggest that support for medicalization also exists and could be significant among women originating from low-medicalized contexts, such as Burkina Faso. This underscores the need for interventions that also take into account the differences among migrant communities and their potential requests in this regard. Moreover, our data show that among women who favor FGM/C continuation, those who condition their support on the grounds of its medicalization are around 38%. The results are similar when considering only cut women, suggesting that migrant women in Italy are more likely to abandon the practice rather than turn to medicalization.

This study also aims to elucidate the possible correlates of support for the choice of medicalization (RQ2). Factors such as education, personal experience of FGM/C, and originating from a country where medicalization is more prevalent are positively correlated with support for medicalization. In particular, our findings document a noteworthy correlation between higher education levels (secondary and tertiary) and support for medicalization as opposed to unconditional support. This challenges the conventional notion that education may lead to less support for FGM/C practices [36]. Instead, it suggests that education may open new avenues of thought, potentially influencing younger women to consider medicalized forms. This complex relationship between education and FGM/C warrants further exploration. The profile of these women seems that of individuals who, despite a higher educational level, do not reject the symbolic significance of the practice but, at the same time, do not underestimate its health risks. As expected, the diffusion of medicalization in the country of origin is positively correlated with support for medicalization. Moreover, support for medicalization is lower among women with a higher number of daughters, suggesting that this support might be more idealized and tends to decrease when the issue of performing FGM/C on daughters becomes more concrete. The same applies to women living apart from their partner, who might not have faced the issue of cutting a daughter. When considering women who do not support the practice as the reference, more traditional indicators known for explaining support for FGM/C become significant. These include experiences of FGM/C, which are positively correlated to support for medicalization, higher age at migration (a proxy for longer socialization in the country of origin), and younger age, again suggesting a possible idealization of medicalized FGM/C among younger women.

The perceived benefits attached to FGM/C **(RQ3)** represent a further element worthy of attention. The most important reasons cited among those who condition their support to medicalization are responding to religious requests, better hygiene of female genitalia, and enhancing marriageability.

Although our study covers a critical topic largely unexplored in Western countries and definitely unreported in Italy, it has some primarily data-driven limitations. First, the analysis is based only on a single country of migration and may not be fully generalizable to other contexts. Second, despite an ad hoc survey design, these data share most of the expected

limitations of surveys on hard-to-reach populations [51] and surveys based on self-reported data on FGM/C status that may be affected by under-reporting of the experience of FGM/C or for the support of its continuation driven by social desirability [53,54]. While support for medicalized FGM/C may be just as susceptible to underreporting as unconditional support for FGM/C, there is a possibility that some women might conditionally express their support for medicalization due to social desirability, even if they actually support the practice unconditionally. However, women who are particularly inclined to provide socially desirable answers are more likely to conceal their support rather than condition it on medicalization. While expressing support for abandonment may disclose hidden support for the practice, we think that medicalization may be less biased as a choice. There is no solid ground to affirm that social desirability may lead women to declare support for medicalization while they could more easily declare they are against the practice. Finally, 73 women who either refused to disclose their opinion on the continuation of FGM/C or declared themselves unsure (6.9% of women who reported having heard of FGM/C) were excluded from the analysis.

Nonetheless, our results yield some interesting findings that highlight the complexity of the practice of FGM/C and underscore the need for more careful consideration when designing contrasting policies. By providing insights into the factors influencing women's preferences for medicalized forms of FGM/C, our study indeed challenges the role of education in eradicating the practice. Furthermore, evidence from this survey, when read through the lens of current literature on the topic, can offer valuable insights for identifying appropriate law enforcement policies with regard to different attitudes.

Recent literature on migrants indicates that women who are against the practice are unlikely to change their stance in favor of it [43]. At the opposite end of the spectrum, the unconditionally favorable minority of women might remain so or, over time, move toward medicalization or abandonment. The first group might be instrumental in combating the practice within their communities, while the second group has to be addressed, with special attention also to women seeking medicalized FGM/C for their children.

Consequently, awareness-raising policies should consider different strategies that exclude medicalization as an option, in collaboration with healthcare providers who view it as "damage reduction." Finally, a return toward unconditional support for FGM/C from women who are in favor of FGM/C, as long as it is medicalized, is not expected. However, because of their profile, medicalization cannot necessarily be considered an intermediate step toward abandonment.

Above all, our data indicate the need for culturally sensitive interventions and comprehensive healthcare education to address the issue of medicalization. There is indeed a potential demand for FGM/C in a medicalized form that could be addressed to health practitioners sharing the same migrant background. Accordingly, specific training should be envisaged in order to better equip them with *ad hoc* competencies. Concurrently, our data suggest that awareness campaigns on medicalized forms of FGM/C might be designed to meet a target of more educated (especially secondary education level) women. Ad hoc intervention studies could carefully investigate this.

In fact, only by understanding the motivations underpinning medicalization and the characteristics of those women who endorse such a practice can policymakers and healthcare providers work towards developing more effective strategies for its prevention.

## Supporting information

**S1 Appendix. Table A1. Multinomial probit choice models. Coefficients and robust standard errors from Model 1a and Model 2a.**
(DOCX)

**S2 Appendix. Table A2. Multinomial probit choice models. Coefficients and robust standard errors from Model 1b and Model 2b.**
(DOCX)

## Author contributions

**Conceptualization:** Livia Elisa Ortensi, Daniela Carrillo.

**Formal analysis:** Livia Elisa Ortensi, Enrico Ripamonti.

**Funding acquisition:** Livia Elisa Ortensi, Patrizia Farina.

**Investigation:** Livia Elisa Ortensi, Patrizia Farina.

**Methodology:** Livia Elisa Ortensi.

**Supervision:** Livia Elisa Ortensi.

**Writing – original draft:** Livia Elisa Ortensi, Patrizia Farina, Daniela Carrillo.

**Writing – review & editing:** Enrico Ripamonti.

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
