## [Decision Letter · Decision Letter 0]

23 Jul 2024

PONE-D-24-11516Exploring support for medicalised female genital mutilation/cutting. A study on migrant women living in Italy (full title)PLOS ONE

Dear Dr. Ortensi,

Thank you for submitting your manuscript to PLOS ONE. After careful consideration, we feel that it has merit but does not fully meet PLOS ONE’s publication criteria as it currently stands. Therefore, we invite you to submit a revised version of the manuscript that addresses the points raised during the review process.

We look forward to receiving your revised manuscript.

Kind regards,

Joyce Jebet Cheptum

Academic Editor

PLOS ONE

Journal Requirements:

3. We notice that your supplementary figures are uploaded with the file type 'Figure'. Please amend the file type to 'Supporting Information'. Please ensure that each Supporting Information file has a legend listed in the manuscript after the references list.

Reviewers' comments:

Reviewer's Responses to Questions

**Comments to the Author**

1. Is the manuscript technically sound, and do the data support the conclusions?

Reviewer #1: Yes

Reviewer #2: Partly

Reviewer #3: Yes

2. Has the statistical analysis been performed appropriately and rigorously? 

Reviewer #1: Yes

Reviewer #2: Yes

Reviewer #3: Yes

3. Have the authors made all data underlying the findings in their manuscript fully available?

Reviewer #1: Yes

Reviewer #2: Yes

Reviewer #3: No

4. Is the manuscript presented in an intelligible fashion and written in standard English?

Reviewer #1: Yes

Reviewer #2: Yes

Reviewer #3: Yes

5. Review Comments to the Author

Reviewer #1: The authors have conducted the research in a rigorous manner. The dat depicted in the findings section supports the final conclusion and recommendations. The findings will definitely add to the knowledge base of the subject area. The authors have good command of English language which makes it easy to read and understand.

Reviewer #2: It would be appropriate to use references that are not more than 5 years to make the study more current. under the Methodology section, please organize it well under Methods and Materials and let there be clarity with the step by step ways that was used to achieve the results. otherwise replicability is not possible. You are trying to analyze an already qualitative study quantitatively and that should come with a step by step clarity.

Please add the qualitative data as a supplementary file or add the link where that can be assessed.

Reviewer #3: A research study was conducted which explores how migrant women from countries where FGM/C is practiced, living in Italy, view the medicalization of FGM/C and the traits of those who endorse this practice. The findings suggest that compared to women who unconditionally support the practice, those who advocate for medicalisation tend to have higher levels of education, be older, be in relationships, and come from countries where FGM/C is more frequently medicalised. Perceived advantages associated with greater endorsement of FGM/C medicalisation include religious sanction, improved marriage opportunities, hygiene considerations, and adherence to traditional cultural norms.

Minor revisions:

1- Table 2: Include the standard deviations that correspond to the means provided in this table.

2- Table 6: Indicate the meaning of the triple asterisks.

6. PLOS authors have the option to publish the peer review history of their article (what does this mean? ). If published, this will include your full peer review and any attached files.

**Do you want your identity to be public for this peer review?** For information about this choice, including consent withdrawal, please see our Privacy Policy .

Reviewer #1: No

Reviewer #2: No

Reviewer #3: No

---

## [Author Response · Author response to Decision Letter 1]

1 Aug 2024

Dear Editor and reviewers,

Thank you for giving us the opportunity to carry out the minor revisions requested for publication in PLOS One.

We met the suggestions when possible and details are reported below (in red) as point by-point answers to observations.

Kind Regards

The Authors

[A word file version of the this letter was uploaded with this re-submission]

Review Comments to the Author

Reviewer #1: The authors have conducted the research in a rigorous manner. The data depicted in the findings section supports the final conclusion and recommendations. The findings will definitely add to the knowledge base of the subject area. The authors have good command of English language which makes it easy to read and understand.

Reviewer #2: It would be appropriate to use references that are not more than 5 years to make the study more current.

Thank you for the suggestions. We have updated older references whenever possible. However, in some instances, we were unable to replace the cited references with more recent findings. As a result, we have retained these references in the text as they are still relevant for framing the study.

under the Methodology section, please organize it well under Methods and Materials and let there be clarity with the step by step ways that was used to achieve the results. otherwise replicability is not possible. You are trying to analyze an already qualitative study quantitatively and that should come with a step by step clarity.

Please add the qualitative data as a supplementary file or add the link where that can be assessed.

We hope we have clearly understood the reviewer's request. We have renamed the section to "Methods and Materials". We also specified that the selected model required us to reshape the original data into long form, and no other steps were carried out on the original data for the analysis. The rest of the analysis was carried out in Stata using cm and cmmprobit commands along with standard post-estimation commands.

We also want to clarify here that our data are fully quantitative, and no qualitative data (mentioned above) were produced at any stage. Data for the study comes from the 2016 national survey on Female Genital Mutilation conducted in Italy as a part of the Daphne project FGM-Prev, which is a quantitative study. The study received ethical clearance from the ethical committee of the University of Milan-Bicocca. The survey was conducted from June to December 2016, covering a representative sample of 1,378 women aged 18+ living in Italy and born in Nigeria, Egypt, Eritrea, Senegal, Burkina Faso, Somalia and the Ivory Coast. Details regarding the data have been updated as per the Journal's checklist. We are open to sharing the codes; however, as explained in the submission, the data are not publicly available.

Reviewer #3: A research study was conducted which explores how migrant women from countries where FGM/C is practiced, living in Italy, view the medicalization of FGM/C and the traits of those who endorse this practice. The findings suggest that compared to women who unconditionally support the practice, those who advocate for medicalisation tend to have higher levels of education, be older, be in relationships, and come from countries where FGM/C is more frequently medicalised. Perceived advantages associated with greater endorsement of FGM/C medicalisation include religious sanction, improved marriage opportunities, hygiene considerations, and adherence to traditional cultural norms.

Minor revisions:

1- Table 2: Include the standard deviations that correspond to the means provided in this table.

Added

2- Table 6: Indicate the meaning of the triple asterisks.

Added

---

## [Decision Letter · Decision Letter 1]

4 Dec 2024

PONE-D-24-11516R1Exploring support for Medicalised Female Genital Mutilation/Cutting. A study on migrant women living in ItalyPLOS ONE

Dear Dr. Ortensi,

Thank you for submitting your manuscript to PLOS ONE. After careful consideration, we feel that it has merit but does not fully meet PLOS ONE’s publication criteria as it currently stands. Therefore, we invite you to submit a revised version of the manuscript that addresses the points raised during the review process. The revised manuscript has been assessed and the reviewers have provided their comments which are available below. Some additional revisions have been requested including further detail in the methodology and further discussion of the study limitations. Please address all of the outlined concerns in a revised version of the manuscript.

We look forward to receiving your revised manuscript.

Kind regards,

Emma Campbell, Ph.D

Staff Editor

PLOS ONE

Reviewers' comments:

Reviewer's Responses to Questions

**Comments to the Author**

1. If the authors have adequately addressed your comments raised in a previous round of review and you feel that this manuscript is now acceptable for publication, you may indicate that here to bypass the “Comments to the Author” section, enter your conflict of interest statement in the “Confidential to Editor” section, and submit your "Accept" recommendation.

Reviewer #1: (No Response)

Reviewer #2: All comments have been addressed

Reviewer #3: (No Response)

Reviewer #4: All comments have been addressed

Reviewer #5: (No Response)

2. Is the manuscript technically sound, and do the data support the conclusions?

Reviewer #1: Yes

Reviewer #2: Partly

Reviewer #3: Yes

Reviewer #4: Yes

Reviewer #5: Yes

3. Has the statistical analysis been performed appropriately and rigorously? 

Reviewer #1: Yes

Reviewer #2: Yes

Reviewer #3: Yes

Reviewer #4: Yes

Reviewer #5: Yes

4. Have the authors made all data underlying the findings in their manuscript fully available?

Reviewer #1: Yes

Reviewer #2: Yes

Reviewer #3: Yes

Reviewer #4: Yes

Reviewer #5: Yes

5. Is the manuscript presented in an intelligible fashion and written in standard English?

Reviewer #1: Yes

Reviewer #2: (No Response)

Reviewer #3: Yes

Reviewer #4: Yes

Reviewer #5: Yes

6. Review Comments to the Author

Reviewer #1: The authors work is carried out in a rigorous manner. The conclusions and recommendations are well supported by the findings of the study. This work has a great potential to add to the scientific knowledge in this field.

Reviewer #2: Line 99: 'At least to our knowledge' please delete this phrase and use a more appropriate scholarly phrase.

line 121-276: Please, fuse/merge the important points with the Introduction, or better separate it as as a review paper.

Instead of 'We' and 'Our' use this study. For example, line 334 can read 'The dependent variables was considered based on this question':

Please engage an English editor.

Lines 347-373; please put it in a table.

line 506-'Our" knowledge' please delete

311-320: Please attach as a supplementary

Reviewer #3: Minor revisions:

1- Table 2: For data summarized as percentages, add the corresponding sample size.

2- Table 6: Replace the "***" in the table with "<.001" which is less confusing since all p-values are <.001.

3- Several tables: P-values never equal zero. Express small p-values as <.001.

Reviewer #4: I would like to congratulate the authors for directing attention to the voices of small but brave respondents who expressed their support for medicalisation of FGM/C. No doubt this has a lot to offer in further exploration of the under-explored motivations for this support. This study though completed in Italy, is an eye opener for other destination countries for migrants such as Australia, UK and USA to invest in similar revisit of existing data or conduct new research.

It was a pleasure reviewing this revised version of the manuscript.

Best wishes.

Reviewer #5: I would like to congratulate the authors, this is truly fascinating paper based on a unique data set which significantly advances our understanding of attitudes to medicalized FGMC among migrant women in Italy. This paper will certainly be well cited, as it provides unique and important details on attitudes and needs of women who have migrated to Europe from high FGMC contexts, but also provides some insights into how attitudes may change overtime, towards medicalisation.

In can confirm that the authors have already made all the revisions required by reviewers 1 2, and 3, In addition, I would like to offer the authors’ the following comments and suggestions.

1) The authors report that FGMC support maybe under-reported in their data. In the discussion, the implications of mis-reporting for their main findings should be outlined. Specifically, I’m concerned that mis-reporting may not be equal across their sample, and that there could be a tendency for both under-reporting and even over-reporting among different sub-groups. For example, highly educated women may be more likely to mis-report in some way. A previous study that used both indirect and direct methods found that the most highly educated people are more likely to give socially desirable answers, under-reporting their “true” preferences for FGMC in self-reports (Gibson et al 2018). Is indicating that you prefer medicalized FGMC (which as you say sound more hygienic, lower health risk) a more socially desirable answer in the context of a survey undertaken in Italy? Or can we assume that all mis-reporting is equal across the women included in the survey – I’m not sure it can as its quite a diverse sample. The author should add this to the limitation section of the discussion, particularly re: the potential for educated people to give socially desirable responses to questions on FGMC support which has been shown before.

2) Also in the limitations section of the discussion. The authors may wish to explicitly discuss the implications of removing women who indicated they were “unsure” or refused to answer the survey. The sample may also be biased here. They should also indicate in the Measures section exactly how many women were excluded from the analyses.

3) Methods/ Discrete choice framework – further details of the survey and the DCE framework would be useful in the methods section. How were the items selected, and how were they presented to respondents? A real benefit of the DCE in measuring preferences, is that when designed well it can it mitigates against strategic biases.

4) “Regression towards unbiased favour” p23. I think the authors are making reference to the expectation that FGMC changes occur in one direction – from traditional cutting methods to medicalization to full abandonment. This is an interesting point and I wonder if this is theoretical or has actually been empirically tested? A bit more explanation would be helpful here. My understanding from the qualitative/anthropological literature is that FGMC may emerge 'de novo' or re-emerge under the right circumstances, so that we can’t be certain on of a simple pathways to abandonment.

5) Small points:

a. I suggest moving the bullet-point list of explanatory variables on p13 into a table for ease of reading and improved structure.

b. P15, end of first paragraph. There is a mention of model which controls for country of origin, but is not presented as the authors say they use a proxy for a “better fit”. Perhaps in the interests of transparency, this could be added to the Supporting Information.

Ref

Gibson MA, Gurmu E, Cobo B, Rueda MM, Scott IM (2018) Indirect questioning method reveals hidden support for female genital cutting in South Central Ethiopia. PLOS ONE 13(5): e0193985. https://doi.org/10.1371/journal.pone.0193985

7. PLOS authors have the option to publish the peer review history of their article (what does this mean? ). If published, this will include your full peer review and any attached files.

**Do you want your identity to be public for this peer review?** For information about this choice, including consent withdrawal, please see our Privacy Policy .

Reviewer #1: **Yes: ** Jane W. Kabo

Reviewer #2: No

Reviewer #3: No

Reviewer #4: **Yes: ** Associate Professor Olayide Ogunsiji

Reviewer #5: **Yes: ** Mhairi Gibson

---

## [Author Response · Author response to Decision Letter 2]

27 Jan 2025

Reviewer #1: The authors work is carried out in a rigorous manner. The conclusions and recommendations are well supported by the findings of the study. This work has a great potential to add to the scientific knowledge in this field.

Thank you

Reviewer #2: Line 99: 'At least to our knowledge.' Please delete this phrase and use a more appropriate scholarly phrase.

Done

line 121-276: Please, fuse/merge the important points with the Introduction, or better separate it as as a review paper.

We separated it as subsection in the section “theoretical background”

Instead of 'We' and 'Our' use this study. For example, line 334 can read 'The dependent variables was considered based on this question':

We rephased as suggested

Please engage an English editor.

Done. Please consider that version with track changes refers to the paper before proofreading. Differences with the clean version are only due to language improvement.

Lines 347-373; please put it in a table.

Done, table have been renamed accordingly

line 506-'Our" knowledge' please delete

Done

311-320: Please attach as a supplementary

This part details the methodology and cannot be shown only in the supplementary materials. Also, reviewer 5 asks for more details instead.

Reviewer #3: Minor revisions:

1- Table 2: For data summarized as percentages, add the corresponding sample size.

Added

2- Table 6: Replace the "***" in the table with "<.001" which is less confusing since all p-values are <.001.

Done

3- Several tables: P-values never equal zero. Express small p-values as <.001.

Done, also in the appendix

Reviewer #4: I would like to congratulate the authors for directing attention to the voices of small but brave respondents who expressed their support for medicalisation of FGM/C. No doubt this has a lot to offer in further exploration of the under-explored motivations for this support. This study though completed in Italy, is an eye opener for other destination countries for migrants such as Australia, UK and USA to invest in similar revisit of existing data or conduct new research.

It was a pleasure reviewing this revised version of the manuscript.

Best wishes.

Thank you

Reviewer #5: I would like to congratulate the authors, this is truly fascinating paper based on a unique data set which significantly advances our understanding of attitudes to medicalized FGMC among migrant women in Italy. This paper will certainly be well cited, as it provides unique and important details on attitudes and needs of women who have migrated to Europe from high FGMC contexts, but also provides some insights into how attitudes may change overtime, towards medicalisation.

In can confirm that the authors have already made all the revisions required by reviewers 1 2, and 3, In addition, I would like to offer the authors’ the following comments and suggestions.

1) The authors report that FGMC support maybe under-reported in their data. In the discussion, the implications of mis-reporting for their main findings should be outlined. Specifically, I’m concerned that mis-reporting may not be equal across their sample, and that there could be a tendency for both under-reporting and even over-reporting among different sub-groups. For example, highly educated women may be more likely to mis-report in some way. A previous study that used both indirect and direct methods found that the most highly educated people are more likely to give socially desirable answers, under-reporting their “true” preferences for FGMC in self-reports (Gibson et al 2018).

Thank you for this crucial observation. As in most FGM/C surveys, it is not possible to control for misreport of FGM/C status or support (the cited text is a notable exception). We add this reference to the discussion.

Is indicating that you prefer medicalized FGMC (which as you say sound more hygienic, lower health risk) a more socially desirable answer in the context of a survey undertaken in Italy? Or can we assume that all mis-reporting is equal across the women included in the survey – I’m not sure it can as its quite a diverse sample. The author should add this to the limitation section of the discussion, particularly re: the potential for educated people to give socially desirable responses to questions on FGMC support which has been shown before.

Thank you for this crucial observation. As in most FGM/C surveys, it is not possible to control for misreport of FGM/C status or support (the cited text is a notable exception). We add this reference to the discussion. However we do not have solid ground to affirm that social desirability may lead women to declare support for medicalization while they could more easily declare they are against the practice. While reporting support for abandonment may disclose hidden support for the practice, we think that medicalization may be less biased as a choice.

This is what we added to the text “these data share most of the expected limitations of surveys on hard-to-reach populations [51] and surveys based on self-reported data on FGM/C status that may be affected by under-reporting of the experience of FGM or for the support of its continuation driven by social desirability [53, 54]. While support for medicalised FGM/C may be as affected by underreporting as unbiased support for FGM/C, there is the possibility that some women may declare a support conditioned to medicalisation due social desirability, while they unconditionally support the practice. However, women particularly oriented towards socially desirable answers are expected to opt for disclosing their support rather than conditioning it to medicalisation.”

2) Also in the limitations section of the discussion. The authors may wish to explicitly discuss the implications of removing women who indicated they were “unsure” or refused to answer the survey. The sample may also be biased here. They should also indicate in the Measures section exactly how many women were excluded from the analyses.

Thank you for your request, which allowed us to disentangle the issue. 57 women are excluded as they declared they had never heard of FGM/C (4.1%). For this reason, they were not asked about their support or intention. Among those who have heard of FGM/C 16 (1.2%) refused to answer and 57 (4.1%) were unsure. Those who are unsure or refuse to answer are excluded because we cannot attribute to their choice any FGM/C-choice related characteristic. However, we lose for this reason only 6.9% of the women aware of FGM/C.

3) Methods/ Discrete choice framework – further details of the survey and the DCE framework would be useful in the methods section. How were the items selected, and how were they presented to respondents? A real benefit of the DCE in measuring preferences, is that when designed well it can it mitigates against strategic biases.

The choice derives from the answer to the item “Do you think that this practice [FGM/C] should continue?”. Possible answers were “No”, “Yes”, “Yes but only if carried out in a hospital”, “Unsure”, and “Refuses to answer”. Women who agreed with the third option were therefore considered favourable to FGM/C conditionally on medicalisation.

This is explained in the Method section. So choices are framed accordingly. This strategy helps avoiding social desirability bias as it is not framed in terms of intention on the body of their daughter but in terms of an opinion.

4) “Regression towards unbiased favour” p23. I think the authors are making reference to the expectation that FGMC changes occur in one direction – from traditional cutting methods to medicalization to full abandonment. This is an interesting point and I wonder if this is theoretical or has actually been empirically tested? A bit more explanation would be helpful here. My understanding from the qualitative/anthropological literature is that FGMC may emerge 'de novo' or re-emerge under the right circumstances, so that we can’t be certain on of a simple pathways to abandonment.

This is an interesting observation. There is no evidence in the literature on countries of origin or among migrants of a return or new emergence of the practice where it is not present or has been abandoned or of an abandonment of medicalization for unmedicalized practice. Reasons are also discussed in the theoretical section in terms of the relationship with the ideas of safety and modernity. However, we slightly changed the text by adding some references. We also explain that a reason not to expect change from support to medicalization to unbiased medicalisation is due to the profile of women who are more educated and, therefore, are more likely to consider health-related risks.

5) Small points:

a. I suggest moving the bullet-point list of explanatory variables on p13 into a table for ease of reading and improved structure.

Done, also in line with suggestion from reviewer 1

b. P15, end of first paragraph. There is a mention of model which controls for country of origin, but is not presented as the authors say they use a proxy for a “better fit”. Perhaps in the interests of transparency, this could be added to the Supporting Information.

We deleted this sentence, not to add too many materials that are not really useful for the readers. By using the country of origin instead of the prevalence, we lose information about the prevalence of medicalization that is much more informative and relevant for the interpretation of results while having a worst fit.

Gibson MA, Gurmu E, Cobo B, Rueda MM, Scott IM (2018) Indirect questioning method reveals hidden support for female genital cutting in South Central Ethiopia. PLOS ONE 13(5): e0193985. https://doi.org/10.1371/journal.pone.0193985

7. PLOS authors have the option to publish the peer review history of their article (what does this mean?). If published, this will include your full peer review and any attached files.

Do you want your identity to be public for this peer review? For information about this choice, including consent withdrawal, please see our Privacy Policy.

Reviewer #1: Yes: Jane W. Kabo

Reviewer #2: No

Reviewer #3: No

Reviewer #4: Yes: Associate Professor Olayide Ogunsiji

Reviewer #5: Yes: Mhairi Gibson

Addendum in response of the submisssion being re-opened

1. Please upload a copy of Figure 1-3 which you refer to in your text on page 18. Or if the figure is no longer to be included as part of the submission please remove all reference to it within the text.

The figures were uploaded as supporting materials, they have now been uploaded is figures.

2. In the online submission form, you indicated that your data is available only on request from a third party. Please note that your Data Availability Statement is currently missing [the name of the third party contact or institution / contact details for the third party, such as an email address or a link to where data requests can be made]. Please update your statement with the missing information.

---

## [Decision Letter · Decision Letter 2]

12 Mar 2025

PONE-D-24-11516R2Exploring support for medicalized female genital mutilation/cutting. A study on migrant women living in ItalyPLOS ONE

Dear Dr. Ortensi,

Thank you for submitting your manuscript to PLOS ONE. After careful consideration, we feel that it has merit but does not fully meet PLOS ONE’s publication criteria as it currently stands. Therefore, we invite you to submit a revised version of the manuscript that addresses the points raised during the review process.

We look forward to receiving your revised manuscript.

Kind regards,

Achenef Asmamaw Muche, Ph.D.

Academic Editor

PLOS ONE

Journal Requirements:

Reviewers' comments:

Reviewer's Responses to Questions

**Comments to the Author**

1. If the authors have adequately addressed your comments raised in a previous round of review and you feel that this manuscript is now acceptable for publication, you may indicate that here to bypass the “Comments to the Author” section, enter your conflict of interest statement in the “Confidential to Editor” section, and submit your "Accept" recommendation.

Reviewer #3: (No Response)

Reviewer #5: (No Response)

2. Is the manuscript technically sound, and do the data support the conclusions?

Reviewer #3: Yes

Reviewer #5: Yes

3. Has the statistical analysis been performed appropriately and rigorously? 

Reviewer #3: Yes

Reviewer #5: Yes

4. Have the authors made all data underlying the findings in their manuscript fully available?

Reviewer #3: No

Reviewer #5: Yes

5. Is the manuscript presented in an intelligible fashion and written in standard English?

Reviewer #3: Yes

Reviewer #5: (No Response)

6. Review Comments to the Author

Reviewer #3: Minor revisions:

1- Table 2: For data summarized as percentages, add the corresponding sample size.

2- Table 6: Replace the "***" in the table with "<.001" which is less confusing since all

p-values are <.001.

3- Several tables: P-values never equal zero. Express small p-values as <.001.

Reviewer #5: Thank you for addressing the questions and comments constructively. It is a very interesting and worthwhile analyses. In general the manuscript reads well, however I note that some of the additional/clarifying text which has been added to R2 version needs some simple editing/rephrasing to improve clarity.

Line 505 - if I understand correctly, "unbiased support for FGMC" should be "unconditional support for FGMC"?

Line 507 - needs an extra clause or sentence to explain your argument. You express it clearly and concisely in your response to reviewers document - that it is your expectation that the most likely scenario is that women declare they are against the practice or similar. The text inserted in the manuscript and that pasted into the response to reviewer document are slightly different, so it might be worth checking this section carefully.

Line 521 - the statement "while the second group needs to be addressed, considering medicalization an an evolved form of FGMC" is unclear.

Line 522-3, ditto. Maybe replace "which may instead be fuelled by" with "in"

Line 525. "one does not expect a regression to an unbiased favour" needs rewording to improve clarity. I expect unbiased favour could be replaced with unconditional support. I would avoid first person terms like "one", and also the term "regression" for different reasons.

7. PLOS authors have the option to publish the peer review history of their article (what does this mean? ). If published, this will include your full peer review and any attached files.

**Do you want your identity to be public for this peer review?** For information about this choice, including consent withdrawal, please see our Privacy Policy .

Reviewer #3: No

Reviewer #5: No

---

## [Author Response · Author response to Decision Letter 3]

14 Mar 2025

Review Comments to the Author

Reviewer #3: Minor revisions:

1- Table 2: For data summarized as percentages, add the corresponding sample size.

2- Table 6: Replace the "***" in the table with "<.001" which is less confusing since all

p-values are <.001.

3- Several tables: P-values never equal zero. Express small p-values as <.001.

All these comments had already been addressed in the previous round of revision. No additional changes needed to be made.

Please note that Table 2 was renumbered into Table 3 in the previous revision round.

Reviewer #5: Thank you for addressing the questions and comments constructively. It is a very interesting and worthwhile analyses. In general the manuscript reads well, however I note that some of the additional/clarifying text which has been added to R2 version needs some simple editing/rephrasing to improve clarity.

Line 505 - if I understand correctly, "unbiased support for FGMC" should be "unconditional support for FGMC"?

We changed the term “unbiased” into “unconditional” support for better clarity

Line 507 - needs an extra clause or sentence to explain your argument. You express it clearly and concisely in your response to reviewers document - that it is your expectation that the most likely scenario is that women declare they are against the practice or similar. The text inserted in the manuscript and that pasted into the response to reviewer document are slightly different, so it might be worth checking this section carefully.

Thank you. Building on the answer provided in the rebuttal letter you mentioned, we added this phrase in the text: “There is no solid ground to affirm that social desirability may lead women to declare support for medicalization while they could more easily declare they are against the practice”.

Also, the text inserted in the answer differed slightly from the paper because the response letter was prepared before the text was proofread, as requested by another reviewer. Sorry about this.

Line 521 - the statement "while the second group needs to be addressed, considering medicalization an an evolved form of FGMC" is unclear.

We changed into: “The first group might be complicit in combating the practice within their communities, while the second group has to be addressed, with special attention also to women seeking medicalized FGM/C for their children”.

Line 522-3, ditto. Maybe replace "which may instead be fuelled by" with "in"

Done

Line 525. "one does not expect a regression to an unbiased favour" needs rewording to improve clarity. I expect unbiased favour could be replaced with unconditional support. I would avoid first person terms like "one", and also the term "regression" for different reasons.

We changed into: “Finally, a return toward unconditional support for FGM/C from women who are in favor of FGM/C, as long as it is medicalized, is not expected. However, because of their profile, medicalization cannot necessarily be considered an intermediate step toward abandonment”.

For the same reasons, we changed the term “evolve” into “move” in another section of the text.

---

## [Editor Report · Decision Letter 3]

28 Mar 2025

Exploring support for medicalized female genital mutilation/cutting. A study on migrant women living in Italy

PONE-D-24-11516R3

Dear Dr. Ortensi,

We’re pleased to inform you that your manuscript has been judged scientifically suitable for publication and will be formally accepted for publication once it meets all outstanding technical requirements.

Kind regards,

Achenef Asmamaw Muche, Ph.D.

Academic Editor

PLOS ONE
---

## [Editor Report · Acceptance letter]

PONE-D-24-11516R3

PLOS ONE

Dear Dr. Ortensi,

I'm pleased to inform you that your manuscript has been deemed suitable for publication in PLOS ONE. Congratulations! Your manuscript is now being handed over to our production team.

Kind regards,

on behalf of

Dr. Achenef Asmamaw Muche

Academic Editor

PLOS ONE